# Cascaded Transforming Multi-task Networks For Abdominal Biometric Estimation from Ultrasound

**Matthew D. Sinclair[1], Juan Cerrolaza Martinez[1], Emily Skelton[2], Yuanwei Li[1],
Christian F. Baumgartner[3], Wenjia Bai[1], Jacqueline Matthew[2], Caroline L. Knight[2],
Sandra Smith[2], Jo Hajnal[2], Andrew P. King[2], Bernhard Kainz[1], Daniel Rueckert[1]**

**Biomedical Image Analysis Group**
Department of Computing, Imperial College London, UK
{m.sinclair, j.cerrolaza-martinez, yuanwei.li09}@imperial.ac.uk
{b.kainz, d.rueckert}@imperial.ac.uk

**School of Biomedical Engineering and Imaging Sciences**
King's College London, UK
{emily.skelton, jacqueline.matthew, caroline.l.knight}@kcl.ac.uk
{sandra.smith, jo.hajnal, andrew.king}@kcl.ac.uk

**Computer Vision Lab**
ETH Zurich, Switzerland
baumgartner@vision.ee.ethz.ch

## Abstract

Measurement of biometrics from fetal ultrasound (US) images is of key importance in monitoring healthy fetal development. Under the time-constraints of a clinical setting however, accurate measurement of relevant anatomical structures, including abdominal circumference (AC), is subject to large inter-observer variability. To address this, an automated method is proposed to annotate the abdomen in 2D US images and measure AC using a shape-aware, multi-task deep convolutional neural network in a cascaded model framework. The multi-task loss simultaneously optimises both pixel-wise segmentation and shape parameter regression. We also introduce a cascaded shape-based transformation to normalise for position and orientation of the anatomy, improving results further on challenging images. Models were trained using approximately 1700 abdominal images and compared to inter-expert variability on 100 test images. The proposed model performs better than inter-expert variability in terms of mean absolute error for AC measurements (3.95mm vs 5.89mm), and Dice score (0.962 vs 0.955). We also show that on the most challenging test images, the proposed method significantly improves on the baseline model, and inference runs in near real-time which could aid clinical workflow.

## 1 Introduction

Ultrasound (US) is a low-cost, non-ionising imaging modality used for clinical fetal screening. A mid-trimester US scan, typically carried out between 18-22 weeks gestation, is used in most countries as part of standard prenatal care. Specific imaging planes are acquired during the scan of different

1st Conference on Medical Imaging with Deep Learning (MIDL 2018), Amsterdam, The Netherlands.

anatomies, referred to as standard planes. Biometric measurements of the head, abdomen and femur are commonly used to estimate both fetal age and weight from these standard planes.

Pitfalls of US however include acoustic shadow, speckle noise, motion blurring and low signal-to-noise ratio, making the identification of soft tissue and bony landmarks, and subsequent biometric assessment, challenging tasks for sonographers. There is considerable inter-observer variability in the measurement of different anatomical structures due to varying levels of sonographer expertise and directed attention fatigue. Additionally, annotation of the abdomen is subject to greater variability than the head for example, due to lower contrast soft tissue boundaries, shadow artefacts and boundary ambiguities resulting from proximity to other anatomical structures such as the limbs and spine.

## 1.1 Related Work

In recent years, deep convolutional neural networks have demonstrated remarkable performance on visual tasks such as image classification [15] and semantic segmentation [10]. Multi-task learning (MTL) has also been shown to improve performance for networks trained simultaneously on multiple related tasks [8, 5]. MTL updates hidden layers during training to simultaneously perform related tasks, and is effective when the tasks are sufficiently related [12]. Additionally, the use of auto-context schemes has led to improvements in network performance; the prediction probabilities from one network are combined with the associated input image forming a new input for a second network, thus providing additional cues [17, 18]. Appropriate image transformations can also improve network performance, for example with spatial transformer networks [6].

For the task of automatic biometric measurement from standard plane images, a range of non-deep learning methods have been used to estimate head circumference (HC) and biparietal diameter (BPD) [2, 14, 9] as well as abdominal circumference (AC) [2] and femur length [14, 2]. None of these methods have been shown to perform at a human level, nor has their robustness been thoroughly demonstrated on challenging clinical images. More recently, deep convolutional neural networks have been used for segmentation [18] and biometric annotation [11, 7] of the abdomen, as well as the head [16, 18]. Both papers in which an AC measurement method was proposed were restricted by training samples of just 70 [11] and 56 [7] images. Thus resulting Dice scores were considerably lower than those reported in [18] in which a cascaded fully convolutional network (FCN) was trained to perform abdominal segmentation using a training dataset of 900 images. In [16], it was demonstrated that segmentation with an FCN followed by an ellipse fitting step [4] produced human-level performance on biometric estimation of HC and BPD, and generated plausible annotations for even the most challenging test images. As shown in this paper however, applying the same method to measure AC from standard plane abdominal images, while producing state-of-the-art results, produces a number of failure cases on the most challenging test images, which we improve with proposed methods.

## 1.2 Contributions

Given that the target annotation produced by a sonographer to measure AC is an ellipse, it is plausible that a network would benefit from learning to predict ellipse parameters in parallel with semantic segmentation. We hypothesise that training a network to learn both ellipse shape parameters and semantic segmentation of the abdomen will improve the predicted biometric annotation, particularly on more challenging cases where only partial anatomical information is visible due to various artefacts. Furthermore, we hypothesise that due to a non-random distribution of abdominal anatomy orientations and positions in the image data, shape-specific spatial transformations could be suitably used to improve ellipse parameter estimation by providing an initial normalisation of ellipse location and orientation.

In this work, we propose an approach to automatically and robustly annotate abdominal images to measure AC from clinically acquired fetal US data. We propose and compare different deep convolutional neural networks, with the primary improvements to the baseline FCN including: (1) additional convolutional layers in the FCN-8 encoder [10] to increase receptive field for the given task; (2) introduction of an ellipse parameter regression branch for a multi-task loss; (3) use of a cascaded model framework; and (4) introduction of a shape-specific spatial transformation in the cascaded model framework to improve ellipse parameter regression.

We assess the performance of our method by comparing to intra- and inter-expert errors with a 100 patient subset of the test data, demonstrating expert-level performance of the proposed method. The

proposed method outperforms results of other methods reported in the literature [2, 11, 7], additionally showing robustness on challenging cases and performing inference in near real-time.

## 2    Materials

The study population consists of 2,352 2D ultrasound examinations from volunteers at 18-22 weeks gestation, which were acquired and labelled during routine screening by a team of 45 expert sonographers according to UK FASP guidelines [3]. All volunteers gave written informed consent in accordance with ethical committee approval. Eight different ultrasound systems, all of the same make and model (GE Voluson E8), were used to perform the examinations. Each volunteer was examined by a single sonographer, who identified all standard scan planes including the abdominal plane. From the abdominal plane, the sonographer created an annotation to measure the abdominal circumference during the examination using an ellipse tool available on the ultrasound system. Freeze-frame DICOM images of the annotated abdominal plane were saved for all subjects. Screen-capture videos were also saved for the ultrasound examinations of each subject, consisting of, on average, 13 minutes of footage per subject at a frame rate of 30*fps*.

While each examination is performed according to the FASP guidelines, large variability is still introduced by the variations in acoustic shadow artefacts, selected zoom level, fetal anomalies and the position of the fetus, which determines the image content around the abdomen. No special selection was made to remove cases with particular image artefacts or anatomical anomalies from the dataset. Thus our dataset is a good representation of the variety of abdominal images expected in a clinical setting. The mean, minimum and maximum AC measurements in the dataset were 156.8*mm*, 123.0*mm* and 203.7*mm*, respectively.

## 3    Methods

### 3.1    Data Pre-processing

The image preprocessing steps followed are detailed in [16]. Briefly, dashed lines representing ellipse annotations in the freeze-frame DICOM images were extracted with several image processing steps and an ellipse was fitted using the method presented in [4] to provide the ground-truth (GT) abdominal mask. Pixels inside the ellipse were given the value 1, and those outside the ellipse 0. A comparison of the fitted ellipse diameter to that recorded on the US system indicated a match with $< 0.2\%$ error.

To obtain annotation-free images, the videos were parsed for the matching unannotated frame corresponding to each freeze-frame DICOM image. A visual check was performed to ensure all recovered video frames were unannotated and matched the corresponding annotated DICOM image for all subjects. Images were cropped to remove on-screen text and scaled by 0.5 to a size of 320x384 pixels, small enough for efficient training and inference while introducing only a negligible error into the AC measurement. The mean original pixel size was $0.13x0.13mm$, so down-sampling introduced an error of up to about $0.8mm$ in AC measurements, or approximately $0.5\%$ of the mean AC value.

### 3.2    Networks

Deep convolutional neural networks (CNNs) designed for semantic segmentation such as the FCN [10] and U-net [13] architectures learn a set of image filters at multiple spatial scales, producing hierarchical feature maps of increasing coarseness. Further filters then learn to up-sample the coarse feature maps to produce a pixel-wise label prediction at the resolution of the input image. The baseline model used in this study is a FCN with 16 convolutional layers, which was used in [16] producing human-level performance for head annotation. Section 3.2.1 describes the multi-task network combining a segmentation loss and an ellipse parameter regression loss. Section 3.2.3 describes standard and proposed cascaded model frameworks.

#### 3.2.1    Multi-task Network

**Fully Convolutional Network**    Let $x$ be an image and $y$ be its corresponding pixel-wise label map, where a training set $S$ consists of pairs of images and label maps, $S = \{x_i | i = 1, 2, \ldots, N; y_i | i =$

$1, 2, \ldots, N$}. Supervised learning is performed to estimate the network parameters, $\Theta$, to predict label map $y_i$ of image $x_i$ in the training set, by optimising the cross-entropy loss function

$$\min_{\Theta} L_s(\Theta) = -\sum_i \sum_j \log P(y_{i,j}|x_i, \Theta), \qquad (1)$$

where $j$ denotes the pixel index and $P(y_{i,j}|x_i, \Theta)$ denotes the softmax probability produced at pixel $j$ for image (and corresponding label map) $i$. The FCN-8 [10] used as the baseline model has a receptive field of 196x196. An additional stride=2 convolutional layer together with 2 further convolutional layers is added to the FCN-8 to increase the receptive field to 404x404, allowing for the deepest encoder convolutional filters to act on the entire context of the input images.

**Ellipse Parameter Regression**  An ellipse can be parameterised with 5 variables: the radii in the directions of the two axes, $a$ and $b$, the centroid coordinates, $c_x$ and $c_y$, and angle of rotation of axis with radius $a$ from the horizontal plane, $\alpha$. The equation defining the region inside an ellipse is

$$\frac{(\cos\varphi(p_x - c_x) + \sin\varphi(p_y - c_y))^k}{a^2} + \frac{(\sin\varphi(p_x - c_x) + \cos\varphi(p_y - c_y))^k}{b^2} \leq 1, \qquad (2)$$

where $p_x$ and $p_y$ are image pixel coordinates. The GT ellipse parameters were derived from the ellipses fitted to the DICOM annotations, as described in Section 3.1. The parameters $a$, $b$, $c_x$ and $c_y$ are given in terms of pixels, while $\alpha$ is given in radians in the range of $[-\pi/4, \pi/4]$ (we choose that the axis with radius $a$ is more parallel with the horizontal plane than the axis with radius $b$). A squared error (SE) loss is used to regress the variables,

$$\min_{\Theta} L_r(\Theta) = \sum_i (\sigma_i^P - \sigma_i^{GT})^2, \qquad (3)$$

where $\sigma_i^P = \{a, b, c_x, c_y, \alpha\}_i^P$ are the predicted parameters and $\sigma_i^{GT} = \{a, b, c_x, c_y, \alpha\}_i^{GT}$ are the GT parameters of the ellipse in image $i$.

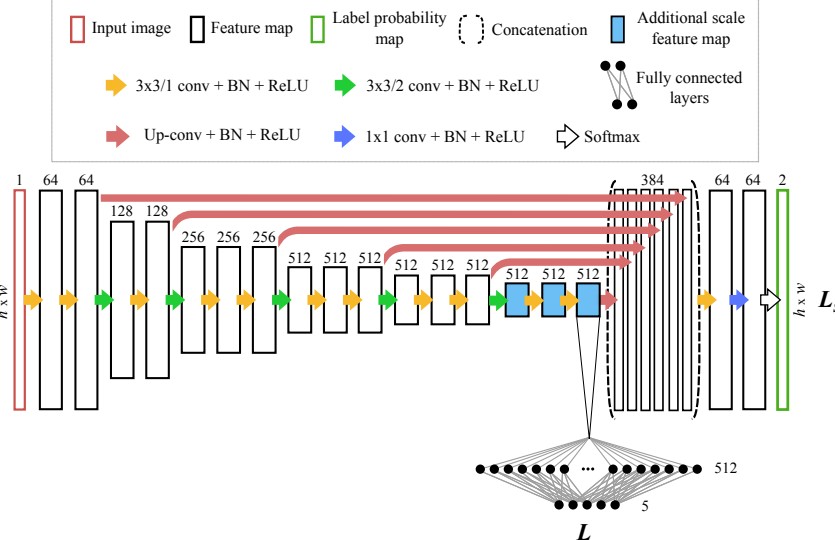

Figure 1: Multi-task network with parameter regression and segmentation branches. In the key above, 3x3/2 = 3x3 filter with stride=2; BN = batch normalisation; ReLU = rectified linear unit activation.

The multi-task loss combining Eq.'s 1 and 3 is simply given by

$$L_m = w_r L_r + w_s L_s, \qquad (4)$$

where $w_r$ and $w_s$ control the weighting of the two losses. The network combining these losses is shown in Fig. 1. Differences to the baseline FCN (which was used in [16]) are: (1) the additional scale of feature maps (blue-filled boxes) which enlarges the receptive field of the associated filters to the whole input image; and (2) the two fully-connected layers for ellipse parameter regression, the input to which is the coarsest set of feature maps.

### 3.2.2 Biometric Estimation

To obtain an annotation, an ellipse is (1) fitted [4] to the segmentation contours of the abdominal region produced by the FCN, referred to as $\text{El}_{seg}$ in the text, and/or (2) retrieved from the regressed ellipse parameters from Eq. 2, referred to as $\text{El}_{par}$ in the text. AC is estimated using the Ramanujan approximation II [1],

$$\text{AC} \approx \pi(a+b)\left(1 + \frac{3h}{10 + \sqrt{4 - 3h}}\right)s_{xy},\tag{5}$$

where $a$ and $b$ are axes radii, $s_{xy}$ is the isotropic pixel size of a given image, and $h = \frac{(a-b)^2}{(a+b)^2}$. This approximation results in an error of $O(h^{10})$ [1], meaning for more circular ellipses like those for the fetal abdomen, the error is negligible.

### 3.2.3 Cascaded Models

Auto-context has recently been used to segment the fetal head and abdomen from 2D US images, demonstrating an improvement over a baseline FCN-8 model [18]. Motivating factors in [18] were to avoid over-fitting on a limited sized dataset ($\approx$ 900 training images) annotated by a single expert, and to overcome prediction boundary ambiguities achieved with the baseline FCN-8 by providing additional cues to cascaded networks. In [16], a considerably larger dataset ($\approx$ 2000 training images) annotated by multiple experts was used to train an FCN-8 model to perform fetal head segmentation, demonstrating human-level performance and results on par with the cascaded model in [18]. Additionally, no mention was made of data augmentation in [18], while in [16] on-the-fly horizontal flipping was employed.

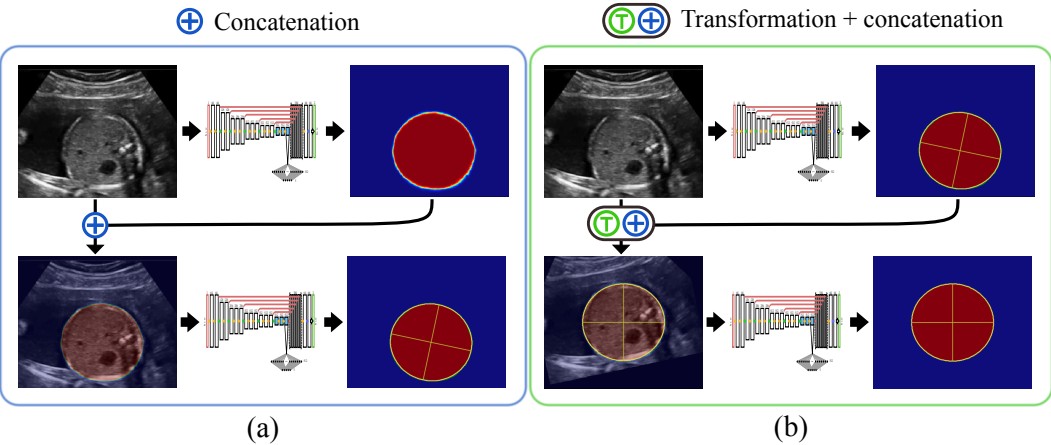

Figure 2: The two cascaded model frameworks: (a) a standard auto-context approach, and (b) the proposed ellipse-based transformation auto-context approach.

Fig. 2 illustrates a standard auto-context framework (a) as well as a proposed framework (b). In the standard auto-context framework, the prediction probability mask, $y_i'$, produced by the first network from input image $x_i$ is concatenated as an additional channel producing a new input $\{x_i, y_i'\}$ to train a second network. Our rationale is that the auto-context approach should aid the direct regression of ellipse parameters by providing a more explicit representation of an ellipse region as an additional input channel to the second network.

In the proposed framework (b) an ellipse is fitted [4] to the segmentation contours produced by the first network. The ellipse parameters are then used to rigidly transform the input image, $x_i$, as well as the output probability map, $y_i'$, from the first network producing a *transformed* input to the second network, $\{x_i^t, y_i'^t\}$, in which the ellipse centroid is centre-aligned with the image, and the ellipse axes are aligned with the image axes (see Fig. 2(b)). Specifically, the axis of radius $a$ is rotated by $\alpha$ (if $a \neq b$) to introduce a minimal image rotation; this avoids significant changes in shadow orientation, which also varies naturally w.r.t. the abdomen depending on abdomen location in the image. Since the regression branch of the network determines 5 parameters based on the whole image, we expect this normalisation will provide a simpler task for the second network's regression branch.

## 4 Experiments

### 4.1 Network Training

The image data was randomly separated into train/test splits of $80\%/20\%$, and then the train data split separated into a train/validation split $90\%/10\%$, resulting in train, test and validation sets of 1698, 466 and 188 images, respectively. Models were trained for a maximum of 100 epochs, or until no improvement on the validation set for 5 epochs in terms of either (a) segmentation Dice or (b) $\text{El}_{par}$ Dice. Networks trained with a multi-task loss used criterion (b), as the parameter regression branch tended to over-fit before the maximum segmentation Dice was achieved, although very little improvement was seen for segmentation Dice after (b) was reached. The Adam optimiser with an initial learning rate of $1e$-5 was used, which was halved for every 3 epochs with no improvement in validation Dice. A mini-batch size of 5 images was used, with on-the-fly data augmentation including random left-right flipping, and small rotations, translations and scaling. Experiments showed that data augmentation was important to maximise performance, particularly for the regression branch.

Several models were trained under different configurations in order to assess the performance of the proposed methods. Namely, experiments were conducted to assess the impact of (1) adding deeper layers in the FCN encoder (prefix: *d*); (2) multi-task loss (prefix: *MT*); (3) standard cascaded framework (prefix: *cas*); (4) cascaded transforming framework (prefix: *tr-cas*). In all experiments, pretrained weights of a VGG-16 classifier trained on ImageNet were used to initialise the weights of the encoder (first 13) layers as this was found to generally improve results. Finally, based on tests, loss weights were set to $w_s = 1.0$ and $w_r = 0.001$ for all experiments with *MT*, while for regression-only experiments $w_s = 0$ and $w_r = 1.0$, and for segmentation-only experiments $w_s = 1.0$ and $w_r = 0$.

### 4.2 Evaluation

Inference was performed on the 466 image test set, and ellipse annotations and AC were obtained as detailed in Section 3.2.2. Mean error (ME) and mean absolute error (MAE) were used to assess AC estimation performance; Dice score was used to assess overlap with GT ellipses; and Hausdorff distance (HDF) was used to assess contour proximity. For the multi-task networks, the two ellipse predictions $\text{El}_{seg}$ and $\text{El}_{par}$ are assessed. Mean and standard deviation (SD) were computed over the whole test set comparing the model-derived ellipses and the GT ellipses in the dataset annotated by the 45 expert sonographers.

In order to further assess the performance of the biometric estimation against human experts, an intra- and inter-observer variability study was performed using 100 images sampled randomly from the test set. Two experts (Expert 1: an engineer with substantial ultrasound experience, and Expert 2: a trained sonographer) used the ellipse tool in MITK[1] to generate two annotations for each image. The images were randomly sampled from the 100-sample set until each was seen twice by both experts. Intra-expert, inter-expert, and model-expert performance was assessed.

Inter-expert error was computed for each image from the differences between each of Expert 1's measurements to each of Expert 2's measurements (i.e. 4 differences per image). Model-expert error was computed from the differences between the model-derived measurement and each expert's measurements (i.e. also 4 differences per image). ME and MAE of the AC measurements were computed along with Dice scores across the 100-sample set.

---

[1]The Medical Imaging Toolkit (MITK), website: mitk.org

Table 1: Model results on all test data, with best metrics shown in bold. $El_{par}$ = ellipse from regressed parameters; $El_{seg}$ = ellipse fitted to segmentation contours; Seg. Dice = segmentation Dice; Dice = $El_{par}$ or $El_{seg}$ Dice. $El_{par}$ and $El_{seg}$ are obtained from separate models in columns FCN and $d$-FCN.

| | FCN | $d$-FCN | $MT$-$d$-FCN | $cas$-$MT$-$d$-FCN | $tr$-$cas$-$MT$-$d$-FCN |
|---|---|---|---|---|---|
| $El_{par}$ | | | | | |
| Dice, % | 91.0±5.5 | 95.0±3.8 | 95.4±3.2 | 96.3±1.9 | **96.4±1.9** |
| HDF, *mm* | 4.72±2.66 | 2.76±1.92 | 2.58±1.63 | **2.07±1.06** | 2.09±1.03 |
| AC ME, *mm* | 0.19±9.66 | 0.88±6.29 | 1.16±5.77 | **0.42±4.48** | 1.29±4.44 |
| AC MAE, *mm* | 7.59±5.98 | 4.47±4.51 | 4.26±4.06 | **3.42±2.91** | 3.49±3.02 |
| $El_{seg}$ | | | | | |
| Seg. Dice, % | 95.7±2.5 | 96.5±1.8 | 96.6±1.7 | **96.7±1.7** | **96.7±1.7** |
| Dice, % | 96.0±2.7 | 96.6±1.9 | 96.7±1.7 | 96.7±1.8 | **96.8±1.7** |
| HDF, *mm* | 2.39±1.82 | 1.94±1.05 | 1.90±1.04 | **1.87±1.00** | 1.93±0.98 |
| AC ME, *mm* | 1.23±5.57 | 0.69±4.71 | 1.01±4.29 | 0.42±4.48 | **0.14±4.26** |
| AC MAE, *mm* | 1.23±5.57 | 3.52±3.21 | 3.53±3.05 | 3.23±2.82 | **3.23±2.80** |

## 5   Results

Table 1 shows the results of the proposed improvements to the baseline FCN on the test dataset. There was a significant improvement in performance between the baseline FCN and all other methods for $El_{seg}$ and $El_{par}$ metrics. Results for $El_{par}$ using a regression-only network (where $w_r = 1$ and $w_s = 0$) are shown in the upper section in the columns titled FCN and $d$-FCN. We observe that the regression-only networks under-perform compared to the segmentation-only networks, but in the multi-task setting they perform more similarly, becoming almost on par with $tr$-$cas$-$MT$-$d$-FCN.

The addition of deeper convolutional layers ($d$) leads to the largest performance increase, while the multi-task loss adds only a small improvement, and the cascaded models boost the $El_{par}$ metrics to be comparable to the $El_{seg}$ metrics without considerably improving $El_{seg}$ metrics. However, even the baseline FCN is able to produce expert-level annotations for most of the test images, since most images do not contain very severe artifacts, thus masking improvements in metrics on the test data.

We therefore examine the effects of the proposed methods on the worst 10% of predictions in terms of $El_{seg}$ Dice produced by the baseline FCN, consisting of 47 test images with a Dice $< 93.8\%$. Fig. 3 shows the improvements in $El_{seg}$ and $El_{par}$ Dice with the proposed models on these test images. Mean±SD for $El_{seg}$ Dice is 90.5±4.0% for the baseline FCN. This increases to 93.4±2.9% with $d$-FCN and improves to 94.0±2.9% with $MT$-$d$-FCN. While the cascaded models do not affect $El_{seg}$ Dice, $El_{par}$ Dice is 90.5±5.5% with $MT$-$d$-FCN, improving to 93.2±3.3% with $cas$-$MT$-$d$-FCN, and 94.2±3.2% with $tr$-$cas$-$MT$-$d$-FCN, on par with $El_{seg}$ Dice for this highest performing model.

Fig. 4 in the Appendix illustrates the annotations produced by the different models on some of the most challenging cases. The poor annotations produced by the baseline FCN (left-most column) are progressively improved with the addition of the proposed model components. Finally, Table 2 shows the results of the intra- and inter-expert variability study in comparison to $cas$-$MT$-$d$-FCN predictions of $El_{seg}$ and $El_{par}$ (which performs similarly to $tr$-$cas$-$MT$-$d$-FCN across the whole test set). The AC error is higher (and Dice lower) for the inter-expert compared to the model-expert results with both $El_{seg}$ and $El_{par}$. Very low intra-expert errors indicate high consistency within experts' repeated annotations. Inter-expert AC MAE and AC ME are also very close, indicating that Expert 1 consistently created smaller ellipses than Expert 2.

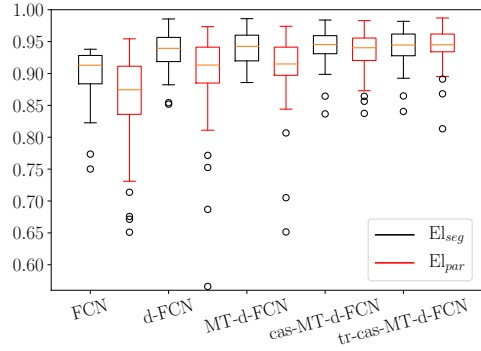

Figure 3: Dice score boxplots of models on the 10% of the worst FCN test cases.

Table 2: Intra- and inter-expert variability results. El$_{seg}$ and El$_{par}$ are predicted from *cas-MT-d*-FCN.

|  | Intra-Expert 1 | Intra-Expert 2 | Inter-Expert | El$_{seg}$-Expert | El$_{par}$-Expert |
|---|---|---|---|---|---|
| AC ME, $mm$ | 0.47±3.09 | 0.46±3.11 | 5.71±2.92 | -0.31±3.46 | 0.21±3.63 |
| AC MAE, $mm$ | 2.43±1.96 | 2.36±2.08 | 5.89±2.61 | 3.95±1.90 | 3.99±1.94 |
| Dice, % | 97.4±1.2 | 97.3±1.3 | 95.5±1.6 | 96.2±1.6 | 96.0±1.5 |

## 6   Discussion and Conclusions

Several improvements to a baseline FCN are introduced for biometric annotation of fetal abdominal images. Firstly, the addition of deeper convolutional layers increased the receptive field of the network to the whole image (and added capacity), boosting performance of El$_{seg}$ and El$_{par}$ metrics. The introduction of a multi-task loss further improved results, particularly on the more challenging test images. The inter-relatedness of the segmentation and regression tasks is self-evident given that the GT mask is derived from an ellipse. The ellipse regression branch of the multi-task network introduces a form of shape regularisation, where the relationship between the size, shape, location and orientation of the ellipse relative to anatomical features in the image is more explicitly enforced, thus regularising the learned filters in the hidden layers shared with the segmentation branch. Similarly, the segmentation branch learns to identify the abdominal region and its boundaries, assisting the regression branch. Compared to the *d*-FCN, the multi-task network *MT-d*-FCN improved boundary localisation in regions of severe shadow artefacts (see bottom two rows of Fig. 4 in Appendix), possibly by using cues from visible boundaries to determine the correct ellipse shape.

The cascaded framework provides the second multi-task network with a more explicit representation of an ellipse. The cascaded transforming network further simplifies the task of the second network by normalising for orientation and location of the ellipse fitted to the initial abdominal segmentation. This produces ellipses from the parameter regression branch with similar accuracy to those generated from the segmentation branch and improves inference for the most challenging test images. The best test data AC MAE, produced from El$_{seg}$ using the *tr-cas-MT-d*-FCN model, is about a quarter of that reported in [2] ($3.2\pm2.8mm$ versus $12.6\pm9.5mm$), and the corresponding Dice score of $96.7\%$ is comparable to the cascaded FCN model in [18], where a Dice score for abdominal segmentation of $96.4\%$ was achieved (although with only a single annotator and for non-ellipse segmentations).

Finally as illustrated in Table 2, the model-expert AC errors and Dice scores produced from both the regression and segmentation branches of the *cas-MT-d*-FCN are better than inter-expert AC errors and Dice scores. The predicted annotations typically lie in between the 2 experts' annotations (both of whom have very low intra-observer errors), and suggests that the network is producing low bias annotations, leveraging the expertise of 45 different sonographers in the training data. Inter- and intra-observer variability from more annotators in future could further corroborate this. Our multi-task framework produces 2 expert-level annotations with an inference frame rate of 30*fps* (or 15*fps* for the cascaded models) on a Titan Xp GPU, and can be used in conjunction with a scan-plane classification model to automatically find and annotate relevant frames[2].

The cascaded transforming multi-task network explicitly normalises image position and orientation w.r.t. a fitted ellipse, although the ellipse axes are not consistently aligned to anatomical features in the abdomen. In future, a comparison could be made to a spatial transformer network which would learn appropriate transformations end-to-end [6]. The terms $w_r$ and $w_s$ could also be weighted automatically using uncertainty as introduced in [8]. A loss between the ellipse parameters generated from the regression and segmentation branches of the multi-task network could be explored too. Finally our models could be applied to other US probe data using fine-tuning or domain adaptation.

**Acknowledgements**   This work was supported by the Wellcome Trust IEH Award [102431, iFind], and by the Department of Health via the National Institute for Health Research (NIHR) comprehensive Biomedical Research Centre award to Guy's & St Thomas' NHS Foundation Trust in partnership with King's College London and King's College Hospital NHS Foundation Trust. We also thank Nvidia Corporation for the donation of the Titan Xp GPU used for this research.

---

[2]see demo: https://www.youtube.com/watch?v=PTSLH6yZWaI

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

## Appendix

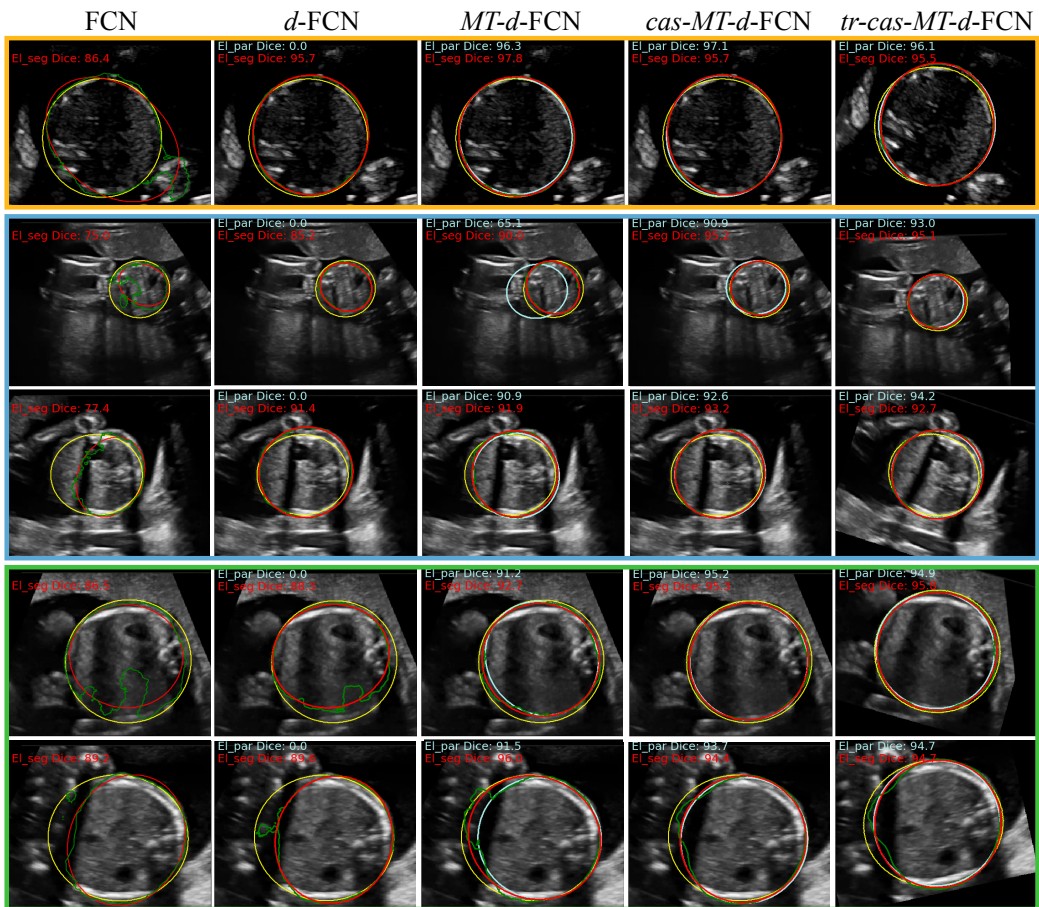

Figure 4: Comparison of results from different models (in columns) on some of the most challenging test cases (one per row). In the images, yellow line = GT annotation; green line = segmentation; red line = $\text{El}_{seg}$; blue line = $\text{El}_{par}$ (drawn only in columns for *MT* models). Orange box (top row): case with low signal-to-noise and ambiguous boundaries - addressed effectively with all proposed models. Blue box (rows 2-3): strong shadow artefacts and low boundary intensities - addressed with *d*-FCN, and improved with subsequent models; green box (rows 4-5): large shadow artefacts - not addressed with *d*-FCN, but addressed with subsequent models. (Zoom to see Dice scores in PDF.)

