# OpenReview forum: "Cascaded Transforming Multi-task Networks For Abdominal Biometric Estimation from Ultrasound"
_MIDL.amsterdam/2018/Conference — MIDL 2018 Poster_

### Review · AnonReviewer3 · 2018-05-09
**Very interesting task, methodology and dataset ; Results are relevant for the more difficult cases,  providing the conclusion that in this case deeper is better.**

**Rating:** 4
**Confidence:** 2

**Review:**

Paper summary:
This work addresses the task of automated segmentation and measurement of fetal abdomen in 2D US images. Authors present a shape-aware multi-task cascaded network to optimize both pixel-wise segmentation and shape parameter regression. The solution is based on the FCN, with 2 additional layers added to reach full image size and measure actual abdomen circumference. An ellipse parametrization regression is added as a multi-task loss and 2 cascaded models are compared, including the case of spatial transformation of one network output based on parameters from the other – for improved ellipse parameter regression.Results are shown on a large dataset with large variability in difficulty levels, with comparison to several experts.

Review:
The paper deals with an important and very challenging task. The concepts proposed for enhancing the networks are very interesting – and seem relevant to the goals of the work. Overall the motivation for the solution is presented clearly, the networks are defined well and many experiments are presented and discussed.

Strengths -
Important clinical task – with initial clinically relevant results.
Results are comparable to expert opinions.
Shows improvement for hard cases when there are artifacts in image.
Paper clearly organized.

Weaknesses –
Table 1 contains a large set of measurements and comparisons – and is somewhat confusing as such. It is suggested that image examples from the Appendix be used in the paper itself – to help clarify the meaning and significance of numbers provided in the Tables. Specifically, the reader can see meaning of image artifacts (different levels of image quality) and the meaning behind the DICE quantities.

Experiments and Results:  It seems that there is minor improvement in the 'normal' cases over the standard FCN.  Actual significance measures are not given.
The concepts specifically focused on in this work seem to help in more rare situations of the more difficult cases. Cascading both methods rarely seem to help.

Work demonstrates that deeper FCN helps in difficult cases (with large image artifacts)  - giving radiologist level performance.
Work can be applied clinically.


**Special Issue:**

Yes

---

### Review · AnonReviewer2 · 2018-05-10
**Very critical clinical problem. Comparing and adapting the state-of-art methods to address the problem with large dataset.**

**Rating:** 3
**Confidence:** 2

**Review:**

Summary:
The paper is addressing the critical clinical problem, abdominal circumference segmentation/estimation, based on classical FCN network with customized addition/adoption. This paper brings in several approaches, combing and comparing them, and presents results based on the large dataset.

Strengths:
Well organized and written
Critical clinical problem
large dataset
good comparison and methods exploration

Weakness:
The author stated in the result section "the addition of deeper convolutional layers (d) leads to the largest performance increase". However, from the table 1, the EIpar can't tell this increase and the EIseg just showed the marginal improvement of Seg. Dice and Dice. How could author draw the stated conclusion? Specifically stated in regards to specifical metrics and discussion of the potential reason might be more convincible.
The author added additional scale layers (3 as shown in figure 1) and stated it enlarges the receptive field. What's the motivation for doing this? Any previous work reported that (appropriate reference will help). Also, provide detail of these layers and how bigger of the receptive fields compared with previous layers. Considering the author said the adding layers brings the most increase in segmentation, an exploration of the best number of adding layers should be also an interesting problem.
The author defined the multi-task loss for optimization. From the figure 1, the Lr loss directly used the input of the adding layers of the fcn. Should be better to show the feature maps of the adding layers.
It's not clear how the author decided the weights of regression loss and segmentation loss. Considering the actual weight of the regression loss is much smaller than segmentation network. How quickly will the training loss converge with and without the regression loss and how different weights will affect the training convergence? Will the regression loss actually help the final task?

**Special Issue:**

Yes

---

### Review · AnonReviewer1 · 2018-05-11
**Measuring abdominal circumference in 2D ultrasound data, evaluated on a large test set**

**Rating:** 3
**Confidence:** 3

**Review:**

This paper describes a method to automatically measure the fetal abdominal circumference in 2D ultrasound images. The authors use their previously published method for head circumference estimation for this task and introduce four improvements to this baseline FCN:
1: add two conv layers to increase the FOV (d-FCN)
2: introduce an ellipse regression parameter (MR-d-FCN)
3: use of a cascade model framework (cas-MT-d-FCN)
4: introduce a shape-specific transformation in the cascade model framework (tr-cas-MT-d-FCN)
The results are evaluated on a test set of 466 images (annotated by one of 45 experts) of which 100 are also annotated by an engineer with ultrasound experience.

Strengths:
The paper is well written
The system is evaluated on a large test set
The system performs similar to an engineer with ultrasound experience and a sonographer

Weaknesses:
- The authors focus in this paper on the four 'improvements' on the baseline FCN, but the used evaluation only shows that the including two conv layers causes the largest improvements, because it ensures that the complete images falls within the FOV, which is methodological not a novel approach.
- It is not valid to examine the effect of the proposed 'improvements' by selecting the worse 10% using the baseline FCN. It is obvious that the baseline FCN shows worse results when you use this method to select the worst 10%. Figure 3 barely shows a difference in ELseg between the other four improvements.  No statistical tests are performed to show significant differences between the d-FCN and the other three 'improvements'. So the major contribution is including two conv layers to increase the FOV.
- The shape-specific transformation both center-aligns and rotates the ellipse major axis to the image x axes. It is questionable if the rotation is a valid procedure, due to two reasons: 1. the anatomical landmarks of the standard plane are not aligned with the minor or major axis. 2. the rotation will lead to rotated shadows, which will never occure in the original ultrasound images. Both these reasons could lead to adding noise instead of adding usefull information. I would recommend to perform the 'tr-cas-MT-d-FCN' without this rotation.
- The authors state that the 'shape-specific transformation' will improve performance of the regression for "orientation, scale and position", but this transformation does not scale the images.
- The authors state in the Discussion: "This indicates that the predicted annotations lie in between the 2 experts’ annotations and suggests that the network has learned to generate annotations with reduced bias based on the 45 sonographers' annotations in the training data". Table 2 shows that the intra-observer variability of both expert 1 and expert 2 is much smaller compared to the inter-observer variability. So, it another explanation for this result could be that the engineer with ultrasound experience consequently over- or under-estimated the AC compared to the sonographer.
- The authors state in the discussion: "There was a significant improvement in performance metrics between the baseline FCN", but not statistical tests are mentioned in the method section. The other results also lack statistical analysis of the results and it is unclear if there is even statistical difference between the d-FCN and the other three 'improvements'
- This paper only focuses on the measurement of the AC when the 2D standard plane is already obtained. It should be mentioned that the acquisition of the standard plane is a major cause of the inter-observer variability and one can discuss if this is even more important than the AC measurement itself.


**Special Issue:**

No

---

### Comment · ~Matthew_David_Sinclair1 · 2018-04-19
**Erratum: inter-observer variability calculations**

Dear readers and reviewers,

Upon revisiting code, an error was found in the calculation of inter-observer errors.

When correcting the way in which mean absolute error (MAE) and mean error (ME) are calculated, three values are changed in Table 2:
(1) Inter-expert AC ME changes from 5.89 +/- 2.61 to 5.71 +/- 2.92
(2) El_seg-expert AC MAE changes from 2.60 +/- 2.30 to 3.95 +/- 1.90
(2) El_par-expert AC MAE changes from 2.77 +/- 2.35 to 3.99 +/- 1.94

This does not change any claims made regarding the improved performance of our proposed model over inter-observer variability, however the text needs to be changed accordingly in the Abstract and last paragraph of the Results section. We will of course make the necessary changes in the final version of the paper if accepted.

---

### Comment · ~Bram_van_Ginneken1 · 2018-05-18
**Selection for longlist for special issue Medical Image Analysis**

Dear authors,

Congratulations on your acceptance to MIDL! We have selected your paper on the longlist for the Medical Image Analysis Special Issue. Please read this page:
https://midl.amsterdam/special-issue-in-medical-image-analysis/
Please answer the three questions that are listed on that page about your interest in submitting to the special issue, potential overlap with other publications, and related publications.

You can post your answer here directly below on openreview.net, or mail me directly at bram.vanginneken@radboudumc.nl.

Best regards, Bram

---

### Decision · Program_Chairs · 2018-05-15
**Paper100 Acceptance Decision**

Poster